# TotalSelfScan: Learning Full-body Avatars from Self-Portrait Videos of Faces, Hands, and Bodies

**Junting Dong**[1][*]    **Qi Fang**[1][*]    **Yudong Guo**[2]    **Sida Peng**[1]    **Qing Shuai**[1]
**Xiaowei Zhou**[1]    **Hujun Bao**[1][†]

[1] Zhejiang University    [2] Image Derivative Inc.

## Abstract

Recent advances in implicit neural representations make it possible to reconstruct a human-body model from a monocular self-rotation video. While previous works present impressive results of human body reconstruction, the quality of reconstructed face and hands are relatively low. The main reason is that the image region occupied by these parts is very small compared to the body. To solve this problem, we propose a new approach named TotalSelfScan, which reconstructs the full-body model from several monocular self-rotation videos that focus on the face, hands, and body, respectively. Compared to recording a single video, this setting has almost no additional cost but provides more details of essential parts. To learn the full-body model, instead of encoding the whole body in a single network, we propose a multi-part representation to model separate parts and then fuse the part-specific observations into a single unified human model. Once learned, the full-body model enables rendering photorealistic free-viewpoint videos under novel human poses. Experiments show that TotalSelfScan can significantly improve the reconstruction and rendering quality on the face and hands compared to the existing methods. The code is available at `https://zju3dv.github.io/TotalSelfScan`.

## 1 Introduction

3D human reconstruction and rendering can be ubiquitously applicable in promising areas such as immersive viewing experiences and telepresence. In most applications for social communications, high-quality face and hand models are essential components. While previous methods [24, 6, 52] demonstrate impressive results of full-body reconstruction, they usually require hundreds of calibrated and synchronized cameras, which makes them impractical to create personalized avatars for general users.

To make human avatar creation more accessible, many recent methods propose to reconstruct the human body model from monocular RGB inputs. Some works [44, 45, 60] propose to reconstruct the human geometry and appearance from a single image by learning pixel-aligned implicit functions. However, relying on the paired 3D data and images for training, these methods have difficulty in generalizing to the in-the-wild images. Some other works propose to reconstruct a person-specific model from a monocular video that records a self-rotation performer holding a fixed pose. The subjects can scan themselves only using a fixed camera without any assistance, which makes the personalized avatar creation possible. For example, VideoAvatar [3] relies on the statistical human model and adds displacements to the vertices of the statistical model for modeling clothing. More

---

[*]The first two authors contributed equally. The authors from Zhejiang University are affiliated with the State Key Lab of CAD&CG.

[†]Corresponding author: Hujun Bao

36th Conference on Neural Information Processing Systems (NeurIPS 2022).

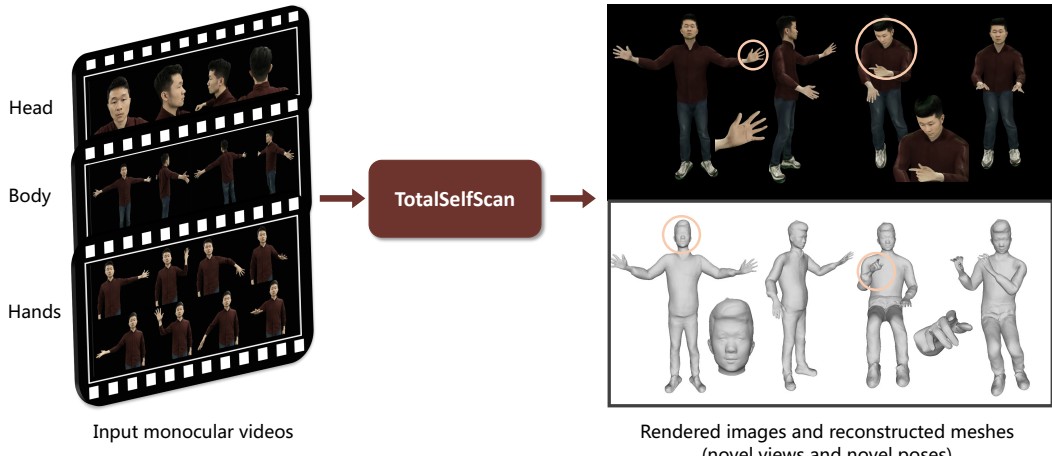

Input monocular videos          Rendered images and reconstructed meshes
(novel views and novel poses)

Figure 1: Given several monocular videos of head, body and hands of a performer, our method is able to reconstruct an animatable full-body avatar of the performer.

recently, SelfRecon [23] proposes to combine the explicit mesh and implicit signed distance field to improve the reconstruction. While these self-rotation video based methods achieve impressive results for the human body, the accuracy of hands and face are relatively low. The main reason is that the image regions occupied by the hands and face are very small in the input self-rotation videos which cover the whole bodies.

In this paper, following the self-rotation video setting, we propose to use a fixed camera to record several monocular videos that focus on the human body, head, and hands respectively, and the performer rotates the part of concern in each video, as shown in Figure 1. Compared to recording a single body video, this setting has almost no additional cost but provides abundant details of essential parts. Given these monocular videos as input, our target is to recover the detailed full-body geometry and appearance, which enables photorealistic rendering of free-viewpoint videos under novel human poses. However, this new task brings many challenges. First, different parts have very different scales, such as the hands and the body, which requires the human model to preserve the details at various scales. Second, each video is part-centric and it is unclear how to seamlessly fuse these videos into a single human model. Third, the appearance of the same part among videos may not be the same due to different lighting. Finally, the self-rotation videos only cover very limited view directions, which makes the rendering under both novel views and novel poses challenging.

To address these problems, we propose a novel multi-part representation to model the body, head, and hands respectively, and fuse the part-specific observations into a single human model. Specifically, we utilize multiple implicit signed distance fields (SDFs) and color fields [57] to represent different human parts in the canonical space. To learn the representation, we propose the part deformation fields to establish the correspondence between each observation space and the canonical space, and the SDF-based volume rendering is utilized to train the model. Then, we propose to fuse the geometry and appearance of adjacent parts to obtain a consistent full-body model. Finally, we extend the ray transformation [58] to the non-rigid case to improve the quality of rendered images under novel human poses.

In summary, this work makes the following contributions:

- We introduce a new task of full-body avatar creation from multiple part-specific videos which provide richer details on human parts than a single video.

- We propose a novel pipeline that reconstructs separate parts from each video and seamlessly fuse them into a unified human model.

- We show that, compared to using a single video, the joint analysis of multiple part-specific videos demonstrates significant reconstruction quality improvement on faces and hands.

## 2 Related work

**Human reconstruction.** Reconstrucing the underlying geometry and appearance of humans has always been an open problem. Previous work can be roughly divided into explicit and implicit representation-based methods. The explicit representation, e.g., a polygon mesh, is obtained in advance in the form of statistical human models [32, 5, 37, 54] or pre-scanned personalized templates [13, 20]. Based on the statistical human model, most works [8, 25, 26, 18, 17, 19] reconstruct the naked body mesh from various inputs and some works further add surface deformation to capture more details [63, 27, 48, 11, 61]. Another line of works [56, 21, 55, 22] utilize personalized templates and deform them by dense non-rigid tracking to achieve performance capture.

The implicit function based methods are prevailing recently. PiFu based methods [44, 45, 60] learn a pixel-aligned implicit function efficiently for both geometry and texture from a single image. However, high-quality 3D models are required for training, which limits their generalization ability. Recently, optimizing a network to represent a person-specific model shows impressive results [14, 46, 10, 41, 39]. Here, we focus on the methods using images as input. Inspired by NeRF [33], Neural Body [41] optimizes the radiance field conditioned on the structured latent codes with only images as supervision. Neural Actor [30] integrates the texture features to enhance the rendering performance. MVP [31] proposes a mixture of volumetric primitives that support efficient rendering of avatars. Furthermore, in order to achieve better animatable effects, learning blend weights automatically from data [39] and incorporating articulated structures [36] are explored. To improve the geometry, [53, 40] represent the human geometry as the signed distance field and use the volume rendering to learn the representation from images. In addition, some works [3, 23] propose to reconstruct the human model from monocular self-rotation videos.

**Total body capture.** Total body capture aims to reconstruct the whole body of the performer including body, face, and hands. Most existing works [32, 42, 28, 9, 7] consider these parts separately. To model the human as a whole, several approaches [24, 38] stitch the part-specific model together and further parameterize the unified model via additional registration and regression. Based on the parametric models, some works propose to optimize the parameters to fit the multi-view [24] or single-view [38, 51] image evidence. To achieve faster inference, some approaches [43, 12, 62, 34] directly regress the parameters of models with neural networks and then integrate or refine them with observations from local regions. Different from the typical mesh representation, imGHUM [4] proposes a generative human model represented by multiple signed distance functions and learns the model from point clouds. A recent work [6] reconstructs full-body avatars based on a disentangled latent space with variational autoencoders conditioned on driving signals, while [52] additionally takes clothing modeling into consideration, which demonstrate impressive reconstruction quality but require hundreds of synchronized high-resolution cameras. In contrast, we reconstruct full-body avatars from monocular videos obtained by a single camera.

## 3 Method

We aim to reconstruct the detailed full-body geometry and appearance from several monocular videos, which enables photorealistic rendering of novel views and novel human poses. Figure 2 shows the overview of the proposed pipeline. We represent the human body as multi-part networks in the canonical space, modeling body, head, and hands, respectively (Section 3.1). To learn the representation from part-specific videos, we first transform the sample points from each observation space to the canonical space via part deformation fields (Section 3.2), and then combine parts into a full-body model (Section 3.3). We train the model with volume rendering (Section 3.4). After training, we adopt the non-rigid ray transformation for novel pose rendering (Section 3.5).

Given four monocular part-specific videos (body, head, and two hands) of the performer, we first utilize the EasyMoCap [1, 16, 15] to estimate SMPL+H [42] parameters for the body and hands videos and utilize an adaptation of [49] to estimate the FLAME [28] parameters for the face video. Then, we adopt [29] to generate the human mask for each frame. In the following, we elaborate each component.

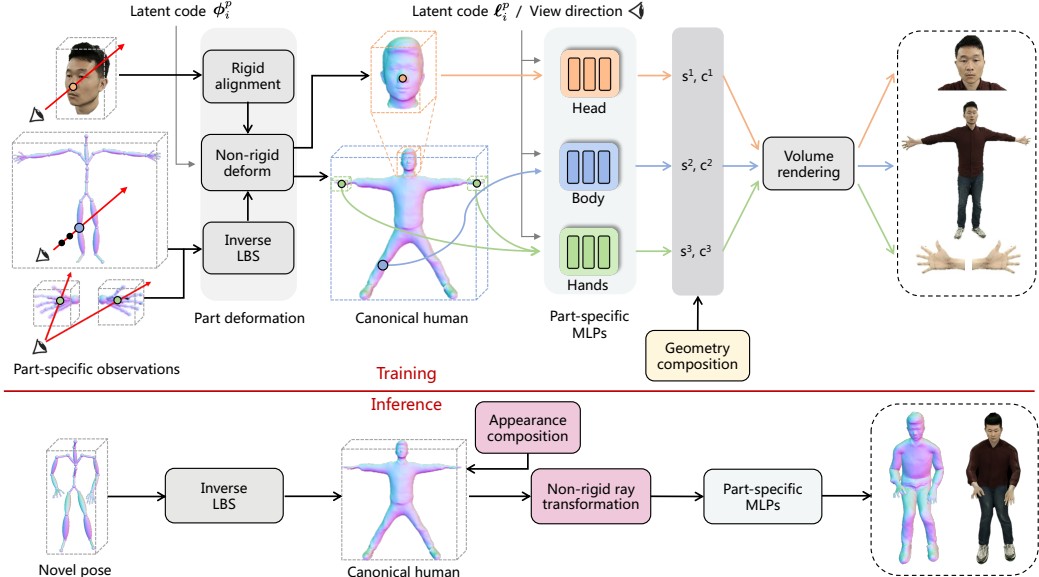

Figure 2: **Overview of the proposed approach.** Given sample points in each part-specific observation space, we transform them to the unified canonical space using part deformations. For each points, the corresponding part network is utilized to predict the signed distance and color and volume rendering is used to synthesize images. To obtain the consistent human geometry, we introduce the geometry composition loss in the training. At inference time, we first composite the appearance of adjacent parts and then animate the canonical human model using input novel poses. To render high-quality images, we adopt the non-rigid ray transformation to replace original view directions with global ones.

## 3.1 Multi-part human model in the canonical space

Similar to [53, 40], the human geometry and appearance are represented as signed distance fields $F_s$ and color fields $F_c$ given by MLP networks. In contrast to previous methods that encode the full human body in a single model, we decompose the human into separate parts (i.e., body, head, and hands) and each part is represented as a single network. Specifically, for the part $p$, the models can be written as follows:

$$s(\mathbf{x}), \mathbf{z}(\mathbf{x}) = F_s^p(\mathbf{x}), \tag{1}$$

$$\mathbf{c}(\mathbf{x}) = F_{\mathbf{c}}^p(\mathbf{x}, \mathbf{z}(\mathbf{x}), \mathbf{n}(\mathbf{x}), \mathbf{d}, \boldsymbol{\ell}_i^p), \tag{2}$$

where $s(\mathbf{x})$ and $\mathbf{c}(\mathbf{x})$ denote the signed distance and color to be decoded at a sampled position $\mathbf{x}$, $\mathbf{d}$, $\mathbf{z}(\mathbf{x})$ and $\mathbf{n}(\mathbf{x})$ denote view direction, the geometry feature and normal in the canonical space, respectively. $\boldsymbol{\ell}_i^p$ denotes the latent code at video frame $i$.

## 3.2 Part deformation

To learn the canonical human model, we need to establish the correspondences between each part-specific observation space and the canonical space. Due to the different properties of each part, we adopt part-specific deformation strategies.

**Body and Hands.** Since the body and hands own a similar articulated structure, we adopt the same deformation strategy consisting of skinning transformation and non-rigid transformation. In particular, for a point $\mathbf{x}$ in the observation space of part $p$ at frame $i$, the corresponding canonical point $\mathbf{x}_c$ can be written as follows:

$$\mathbf{x}_c = T_{ilbs}(\mathbf{x}, \mathbf{p}_i) + T_{nr}^p(T_{ilbs}(\mathbf{x}, \mathbf{p}_i), \boldsymbol{\phi}_i^p), \tag{3}$$

where $T_{ilbs}$ is the standard inverse linear blend skinning algorithm with no learnable parameters and $\mathbf{p}_i$ is the human pose. Note that the blending weights of $\mathbf{x}$ are generated by retrieving the counterpart

of the closest vertex on the template mesh. $T_{nr}^p$ is the part-specific non-rigid displacement field implemented as an MLP network and $\phi_i^p$ is the latent code. Practically, we adopt the SMPL+H model [42] for both body and hands.

**Head.** Different from the articulated structure of the human body, the head is closer to a rigid structure, whose motion can be described as a rigid transformation solved by Structure from Motion [47]. In addition to the rigid transformation, we also introduce the non-rigid transformation similar to the body and hands. Specifically, given a point $\mathbf{x}$ in the observation space at frame $i$, the corresponding point $\mathbf{x}'$ in the head canonical space can be written as:

$$\mathbf{x}' = \mathbf{R}_i\mathbf{x} + \mathbf{T}_i + T_{nr}^p(\mathbf{R}_i\mathbf{x} + \mathbf{T}_i, \phi_i^p), \tag{4}$$

where $\mathbf{R}_i$ and $\mathbf{T}_i$ are the rotation and translation, respectively. To align the head canonical space and the unified canonical space, we register the reconstructed head surfaces $\mathbf{X_{head}}$ and $\mathbf{X_{hbody}}$ obtained from the head and body videos by solving the following optimization problem:

$$\min_{\mathbf{R},\mathbf{T}} \sum_{\mathbf{x}\in\mathbf{X_{head}}} \min_{\mathbf{x}'\in\mathbf{X_{hbody}}} \|(\mathbf{R}\mathbf{x} + \mathbf{T} - \mathbf{x}') \cdot \mathbf{n}'\|_2, \tag{5}$$

where $\mathbf{x}$ and $\mathbf{x}'$ denote the corresponding vertexes of two surfaces, and $\mathbf{n}'$ denotes the normal of the vertex $\mathbf{x}'$. This problem can be solved using the iterative closest point (ICP) algorithm. However, accurate registration requires a good initial alignment. Therefore, to obtain the initial alignment, we first adopt [49] to reconstruct the FLAME model from the head video and then register the FLAME model to the canonical SMPL+H model by solving the following optimization problem:

$$\min_{\mathbf{R},\mathbf{T}} \|\mathbf{R}\mathbf{L_{flame}} + \mathbf{T} - \mathbf{L_{smplh}}\|_2, \tag{6}$$

where $\mathbf{L_{flame}}$ and $\mathbf{L_{smplh}}$ are the pre-defined corresponding landmarks on the FLAME model and the SMPL+H model, respectively.

### 3.3 Part compositions

After warping points from each part-specific observation space to the canonical space, we need to composite separate part models into a unified human model, which contains the compositions of geometry and appearance. Specifically, we define a bounding box for each part in the canonical space and there is overlap between two adjacent bounding boxes. In the bounding box $\mathbf{B}^p$ of part $p$, the corresponding models $F_s^p$ and $F_c^p$ are activated. To generate a realistic unified model, we utilize the following composition strategies in the intersection region of two bounding boxes.

**Geometry composition.** To ensure smooth surface transitions between two adjacent part networks, we introduce the following loss function:

$$L_{\mathrm{g}} = \sum_{\mathbf{x}\in\mathcal{X}_{ij}} \|s^{p_i}(\mathbf{x}) - s^{p_j}(\mathbf{x})\|_2 + \sum_{\mathbf{x}\in\mathcal{S}_j} \|s^{p_i}(\mathbf{x})\|_2, \tag{7}$$

where $\mathcal{X}_{ij}$ is the set of sample points in the intersection region between part $i$ and part $j$, and $\mathcal{S}_j$ is the set of sample points on the zero-level-set of part $j$ in the intersection region. The first term enforces the two signed distance fields to be consistent and the second term further enforces the consistency on the reconstructed surface.

**Appearance composition.** Different from the consistent human geometry across part-specific videos, the appearances of the same part in different videos are usually inconsistent due to uneven and varying lighting conditions. As a result, the learned separate part models generate inconsistent appearances at the same position. To solve this problem, we select the body model as the reference and optimize the appearance code of other parts $\ell^p$ to achieve appearance consistency, which can be written as follows:

$$\min_{\ell^p} \sum_{r\in\mathcal{R}} \|\tilde{\mathbf{C}}^{body}(\mathbf{r}, \ell^{body}) - \tilde{\mathbf{C}}^p(\mathbf{r}, \ell^p)\|_2, \tag{8}$$

where $\mathcal{R}$ denotes the set of rays of sampled intersection region, and $\tilde{\mathbf{C}}^p(\mathbf{r}, \ell^p)$ denotes the rendered pixel color of part $p$. After the appearance code optimization, the appearances between two parts become similar but not exactly the same, which leads to an unwanted seam near the boundary.

To generate a realistic appearance transition, we further fuse the two adjacent color fields in the overlapping region as follows:

$$\mathbf{c}(\mathbf{x}) = \mathbf{c}^{p_i}(\mathbf{x})(1 - \frac{d^{p_i}(\mathbf{x})}{d_s}) + \mathbf{c}^{p_j}(\mathbf{x})\frac{d^{p_i}(\mathbf{x})}{d_s}, \tag{9}$$

where $d^{p_i}(\mathbf{x})$ denotes the distance of $\mathbf{x}$ to part $p_i$ and $d_s$ denotes the depth of overlapping region. Note that the appearance composition is performed after training.

### 3.4 Training

To learn the signed distance filed $s(\mathbf{x})$ and color filed $\mathbf{c}(\mathbf{x})$ from images, we leverage the SDF-based volume rendering [57, 50] to synthesize images and compare them with the input images.

For high-quality reconstruction, we introduce a two-stage training strategy. In the first stage, we train each part model using the part-specific video separately and the loss functions are given as follows:

$$L^p = L^p_{\text{rgb}} + L^p_{\text{mask}} + \lambda_1 L^p_{\text{E}} + \lambda^p_2 L^p_{nr}, \tag{10}$$

$$L^p_{\text{rgb}} = \sum_{r \in \mathcal{R}} \|\tilde{\mathbf{C}}(\mathbf{r}) - \mathbf{C}(\mathbf{r})\|_2, \quad L^p_{\text{mask}} = \sum_{r \in \mathcal{R}} \text{BCE}(\text{sigmoid}(-\rho s^{\mathbf{r}}), M(\mathbf{r})), \tag{11}$$

$$L^p_{\text{E}} = \sum_{\mathbf{x} \in \mathcal{X}^c} (\|\nabla F^p_s(\mathbf{x})\|_2 - 1)^2, \quad L^p_{nr} = \sum_{\mathbf{x} \in \mathcal{X}^c} \|T^p_{nr}(\mathbf{x}, \phi^p_i)\|_2. \tag{12}$$

The first term $L^p_{\text{rgb}}$ denotes the color loss. The second term $L^p_{\text{mask}}$ denotes the mask loss, where $\rho$ is a gradually increasing hyper-parameter and $M(\mathbf{r})$ is the ground-truth mask value. The third term $L^p_{\text{E}}$ denotes the Eikonal loss, where $\mathcal{X}^c$ is the sampled points in the canonical space. The last term $L^p_{nr}$ denotes the displacement regularization. The $\lambda_1$ and $\lambda^p_2$ are two predefined constants and $\lambda^p_2$ is part-specific.

In the second stage, to obtain a consistent full-body model, we further add geometry composition loss. Instead of training all part models jointly, we only optimize the body model while fixing the other part models in the second stage. This strategy can fuse adjacent signed distance fields smoothly while preserving the part details. The loss function is given as follows:

$$L^{union} = L^{body} + L_g. \tag{13}$$

### 3.5 Non-rigid ray transformation

After training, the canonical human model can be animated and rendered under novel human poses. However, when the rays under novel human poses deviate from the training ray distribution, the color field network $F_c$ produces unexpected artifacts. Inspired by [58] that constructs a ray atlas for a rigid object, we extend it to the non-rigid human. Specifically, we first transform each ray of training frames from the observation space to the canonical space. Then, we extract the human mesh in the canonical space and save all view directions $\mathbf{d}^n_v$ of each vertex $v$ on the human mesh. Based on the saved view directions, we can compute a global view direction $\overline{\mathbf{d}}_v$ as follows:

$$\overline{\mathbf{d}}_v = \frac{1}{N} \sum_n \mathbf{d}^n_v, \tag{14}$$

where $N$ is the number of transformed view directions related to the vertex. Finally, for image synthesis under novel human poses, we replace original view directions related to vertex $v$ with the global one $\overline{\mathbf{d}}_v$ for each ray.

## 4 Experiments

### 4.1 Datasets and metrics

**Datasets.** Since there is no existing dataset for our task, we create two new datasets for evaluation. The first dataset *TotalHuman* consists of four subjects. For each person, we use a fixed camera

Table 1: Results of 3D reconstruction of each part on the *SynTotalHuman* dataset.

| | Head | | Hands | | Total | |
|---|---|---|---|---|---|---|
| | P2S↓ | CD↓ | P2S↓ | CD↓ | P2S↓ | CD↓ |
| NeuralBody [41] | 1.55 | 1.46 | 1.29 | 1.19 | 2.16 | 1.98 |
| AniNeRF [39] | 1.80 | 1.69 | 1.12 | 1.00 | 2.55 | 2.21 |
| AniSDF [40] | 0.76 | 0.91 | 0.79 | 0.75 | 1.90 | 1.97 |
| Ours | **0.59** | **0.82** | **0.55** | **0.52** | **1.84** | **1.89** |

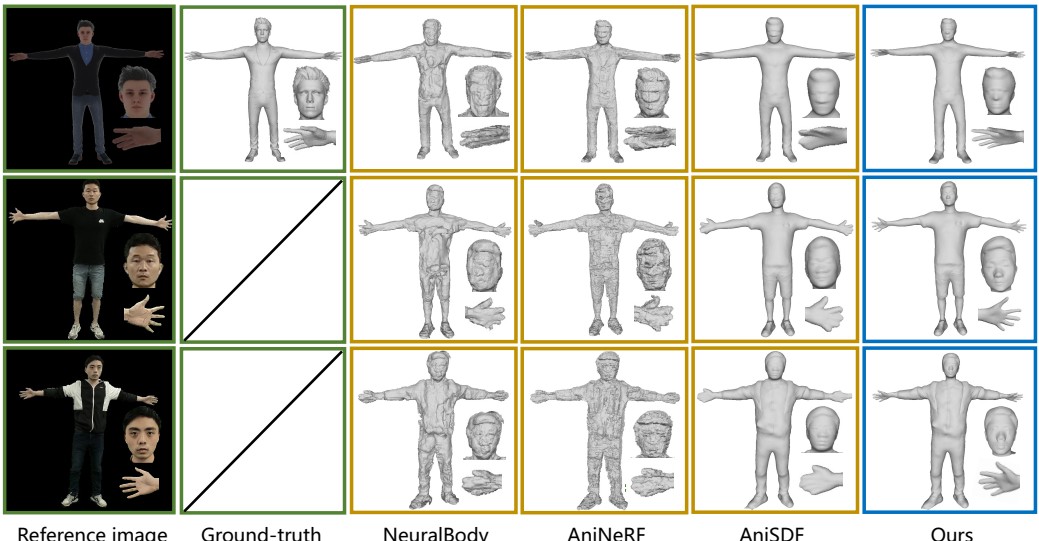

Figure 3: 3D reconstruction on the *SynTotalHuman* and *TotalHuman* datasets.

to record four monocular videos focusing on the body, head, and two hands, respectively. For the body video, the subject turns around while holding a T-pose. For the head video, the subject also self-rotates but the camera focuses on the head. For the two hands, the subject holds a fixed hand pose and moves the hand. We use this dataset for qualitative evaluation.

For quantitative evaluation, we create a synthetic dataset *SynTotalHuman* which contains four animated 3D characters from Mixamo [2]. For each character, similar to *TotalHuman*, we render four monocular part-specific videos for training and render images under novel human poses for evaluation. In addition, we also utilize this dataset to evaluate the accuracy of 3D surface reconstruction.

**Metrics.** For the evaluation of image synthesis, we adopt the following metrics: peak signal-to-noise ratio (PSNR), structural similarity index (SSIM), and learned perceptual image patch similarity (LPIPS) [59]. For 3D reconstruction, we adopt the following two metrics: point-to-surface Euclidean distance (P2S) and Chamfer distance (CD), whose units are both centimeters.

## 4.2 Comparison with the baselines

Since most previous methods only focus on body modeling, we extend the state-of-the-art methods AniNeRF [39] and AniSDF [40] with hands to compare with our method. Since the neural feature field of AniSDF can only converge on small images, we use the color field for image synthesis. We also compare with NeuralBody [41].

**3D reconstruction.** We first compare 3D surface reconstructions of our method and other baselines. In addition to the full-body evaluation, we also compare the reconstruction of the head and hands individually. The quantitative results on the *SynTotalHuman* dataset are shown in Table 1. Thanks to the multi-part representation and the use of part-specific videos, our method outperforms the baselines in all parts, especially the head and hands. We present the qualitative results in Figure 3.

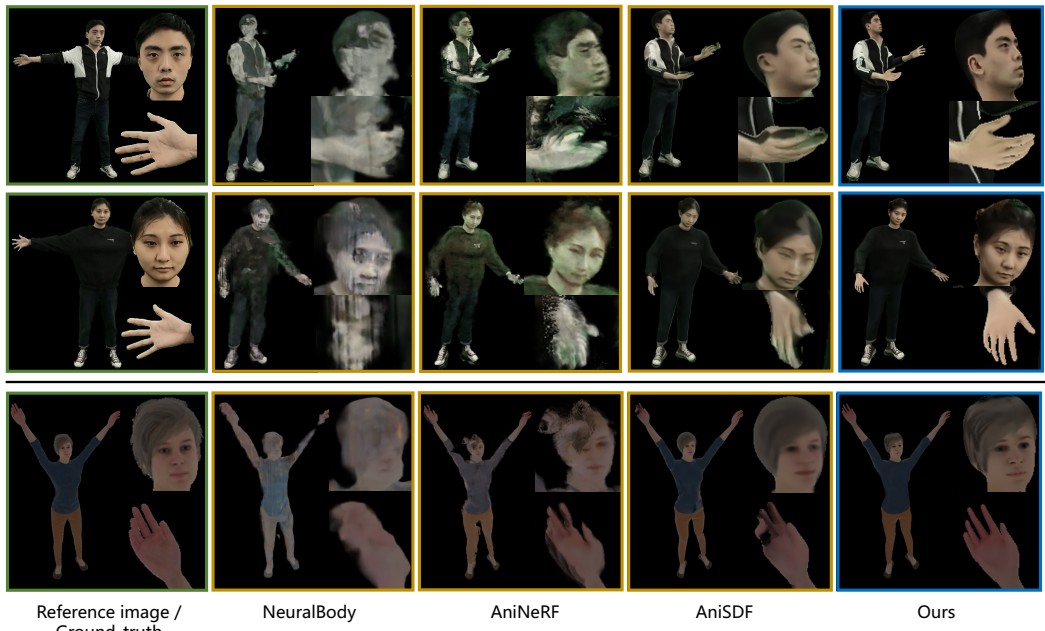

| Reference image / Ground-truth | NeuralBody | AniNeRF | AniSDF | Ours |

Figure 4: Image synthesis under novel human poses on the *TotalHuman* and *SynTotalHuman* datasets. Note that for the *TotalHuman* dataset (first two rows), there are no ground-truth images under novel poses and we put the training images for reference.

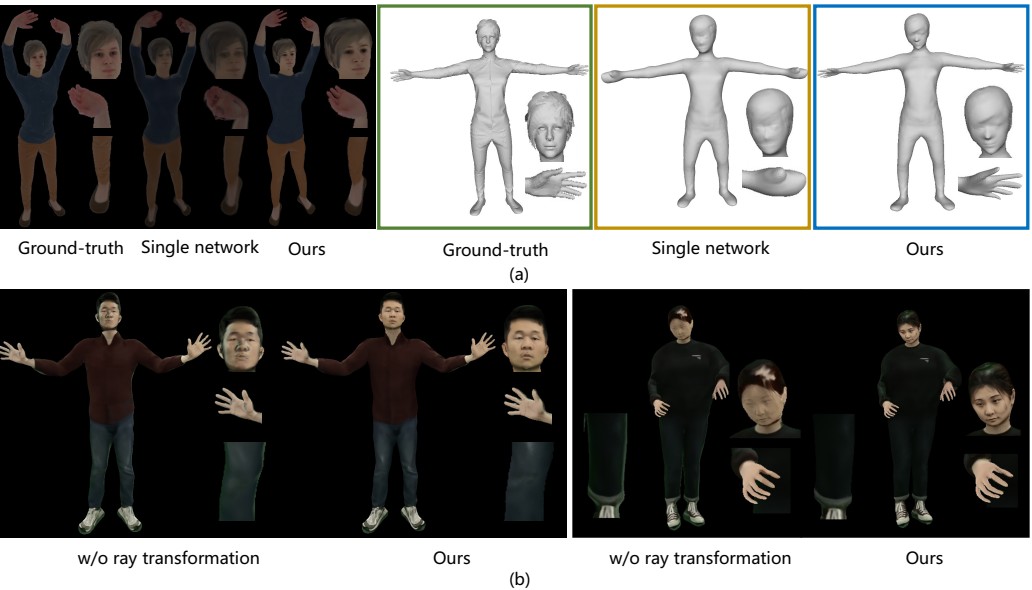

Figure 5: Ablation studies for (a) multi-part networks and (b) non-rigid ray transformation .

**Image synthesis.** We evaluate the image synthesis quality on the *TotalHuman* and *SynTotalHuman* datasets. Specifically, we compare the rendering results of novel human poses under novel views. Similar to the 3D reconstruction, we compare the rendered images of each part. The quantitative results on the *SynTotalHuman* dataset are given in Table 2, which show that our approach achieves the best rendering quality. Figure 4 shows the qualitative results on both datasets. As we can see, the head and hands images rendered by our method significantly outperform the counterparts of the baseline methods.

Table 2: Results of image synthesis under novel human poses of each part on *SynTotalHuman*.

| | Head | | | Hands | | | Total | | |
|---|---|---|---|---|---|---|---|---|---|
| | PSNR↑ | SSIM↑ | LPIPS↓ | PSNR↑ | SSIM↑ | LPIPS↓ | PSNR↑ | SSIM↑ | LPIPS↓ |
| NeuralBody [41] | 18.94 | 0.914 | 0.184 | 18.51 | 0.893 | 0.240 | 19.97 | 0.855 | 0.239 |
| AniNeRF [39] | 20.92 | 0.928 | 0.153 | 20.60 | 0.907 | 0.179 | 23.71 | 0.889 | 0.185 |
| AniSDF [40] | 21.91 | **0.934** | 0.104 | 21.31 | 0.930 | 0.111 | 25.57 | 0.916 | 0.129 |
| Ours | **22.23** | **0.934** | **0.084** | **22.49** | **0.941** | **0.076** | **26.15** | **0.921** | **0.114** |

Table 3: Ablation study on *SynTotalHuman* in reconstruction accuracy.

| | Head | | Hands | | Total | |
|---|---|---|---|---|---|---|
| | P2S↓ | CD↓ | P2S↓ | CD↓ | P2S↓ | CD↓ |
| Single network | 1.03 | 1.26 | 1.25 | 1.22 | 1.95 | 1.97 |
| Ours | **0.59** | **0.82** | **0.55** | **0.52** | **1.84** | **1.89** |

## 4.3 Ablation studies

We conduct the ablation studies to justify the algorithm designs in the proposed method.

**Multi-part networks.** We use multiple networks to represent the different human parts as described in Section 3.1. An alternative is to represent the whole body using a single network which owns the same number of parameters with the multi-part network. The quantitative results of 3D reconstruction and image synthesis are presented in Table 3 and Table 4, respectively. The results show that our multi-part networks outperform the single network by a large margin. We also show the qualitative results in Figure 5 (a).

**Head deformation.** As described in Section 3.2, we first use FLAME-SMPLH registration to initialize the rigid transformation and then refine it with the reconstructed surface alignment. Here, we compare it with the initialized transformation obtained from the FLAME-SMPLH registration. The qualitative results are shown in Figure 6 (a). As we can see, the surface alignment significantly improves the alignment between the head part and the body part.

**Geometry composition.** We introduce the $L_g$ loss in Equation (7) to ensure smooth surface transitions between two adjacent parts. Here, we compare it to the results without this loss. Figure 6 (b) presents the qualitative results on the *TotalHuman* dataset. With the $L_g$ loss, the surface transition between two parts is significantly improved.

**Appearance composition.** For recovering consistent appearance, we first optimize the latent code of head and hand color networks and then fuse the color predictions. To evaluate the proposed method, we compare it with: 1) w/o composition: neither latent code optimization nor color fields fusion is used; 2) w/o color fusion: no color fusion is performed. The qualitative results are shown in Figure 6 (c). As we can see, our method presents much better appearance consistency.

Table 4: Ablation study on the *SynTotalHuman* dataset in image synthesis quality.

| | Head | | | Hands | | | Total | | |
|---|---|---|---|---|---|---|---|---|---|
| | PSNR↑ | SSIM↑ | LPIPS↓ | PSNR↑ | SSIM↑ | LPIPS↓ | PSNR↑ | SSIM↑ | LPIPS↓ |
| single network | 20.38 | 0.919 | 0.120 | 20.75 | 0.927 | 0.140 | 25.19 | 0.907 | 0.144 |
| w/o ray transform | 21.07 | 0.933 | 0.095 | 22.14 | 0.939 | 0.085 | 25.43 | 0.916 | 0.123 |
| Ours | **22.23** | **0.934** | **0.084** | **22.49** | **0.941** | **0.076** | **26.15** | **0.921** | **0.114** |

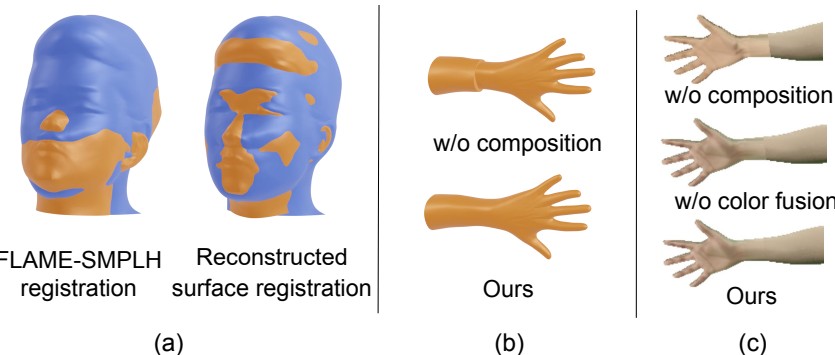

Figure 6: (a) Ablation study for head deformation. The blue mesh is the target mesh reconstructed from the body video. The orange mesh is the source mesh reconstructed from the face video. (b) Ablation study for geometry composition. (c) Ablation study for appearance composition.

**Non-rigid ray transformation.** After training, we introduce the non-rigid ray transformation to improve the novel view synthesis under novel human poses. To analyze its effect, we compare it with the rendering without ray transformation. The quantitative and qualitative results are shown in Table 4 and Figure 5 (b), respectively. The results indicate that our ray transformation greatly improves the rendering quality. Without the ray transformation, there will be severe artifacts on the head, hands, and boundary of the body. The reason is that the self-rotating video covers very limited human poses, and each point on the body is seen from a very limited range of view directions. Therefore, the color field network has difficulty producing high-quality rendering under novel human poses. Replacing the input view directions with global ones is a reasonable way to improve the generalization.

## 5 Limitations

The proposed method has the following limitations. First, the self-rotation human video provides very limited human motions, which makes our method difficult to model pose-dependent deformations. It would be interesting to leverage the existing multi-view human motion datasets to learn a generalizable pose-dependent deformation regressor, which can be conditioned on the human pose and the canonical geometry. Once learned, the generalizable regressor can be applied to new input data or can be further finetuned on the input data to improve the results. Second, our method still needs a relatively long time for training. Recent work [35] uses hash encoding to significantly reduce the training time of implicit functions. Combining this technique into our method is left as future work.

## 6 Conclusion

In this paper, we introduce TotalSelfScan, a convenient approach to creating full-body avatars from several monocular self-rotation videos that focus on the face, hands, and body, respectively. We propose a multi-part network to represent the whole human in the canonical space and the part deformation is utilized to establish the correspondences between the observation frames of each part and the canonical space. We also propose a part composition method to obtain a consistent unified human model. For rendering, we propose the non-rigid ray transformation to render photorealistic free-viewpoint videos under novel human poses. Both quantitative and qualitative results demonstrate the effectiveness of our method to reconstruct high-fidelity avatars from monocular videos. Lastly, using the reconstructed avatars to synthesize unauthorized personal images may have negative societal impact and we strongly discourage such applications.

**Acknowledgements:** This work has been supported by Key Research Project of Zhejiang Lab (No. K2022PG1BB01), NSFC (No. 62172364), and the Information Technology Center and State Key Lab of CAD&CG, Zhejiang University.

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
