# OpenReview forum: "TotalSelfScan: Learning Full-body Avatars from Self-Portrait Videos of Faces, Hands, and Bodies"
_NeurIPS.cc/2022/Conference — NeurIPS 2022 Accept_

### Official Review · Reviewer_Nipv · 2022-07-10

**Rating:** 7
**Confidence:** 4
**Soundness:** 3 good
**Presentation:** 3 good
**Contribution:** 3 good

**Summary:**

This work aims at reconstructing the full-body human from several monocular self-rotation videos including the face, hands, and body videos. Compared to the priors, this work improves the local details significantly with the acceptable cost of several extra part videos and more training time. The experiment results prove the effectiveness of the proposed framework and each introduced component.

**Questions:**

Overall, this work is easy to follow and useful for real human reconstruction applications. I am interested in the prior of pose-dependent deformations discussed in the limitation. I think it is the key limitation of TotalSelfScan when applied to animation. I would expect more discussion on this part.

**Limitations:**

The limitations and potential negative societal impact have been well described.

**Strengths And Weaknesses:**

Strengths

- The proposed multi-part networks can utilize information from different part videos to learn better local details. The ablation study in  Figure5(a) and Table3 show the effectiveness of multi-part networks. I believe part-based data and modeling is the right way to enhance the details for complex objects like humans.

- To connect the observation space and the canonical space, this work first uses part-based deformation fields to transform the sample points from each observation space, and then performs the geometry and appearance composition to generate a unified model.
 Sufficient experiments and comparisons have been done to show the superiority of TotalSelfScan.

- Two new datasets are introduced for part-based model evaluation. I believe they can also benefit the computer vision/graphics community.

- Limitations and reasonable potential future solutions have been well discussed. I believe these can also benefit the following works.

Weaknesses

- The non-rigid ray transformation seems a nice strategy to alleviate the unnatural color artifacts. I would expect more qualitative comparisons and analysis on this part.

---

> ### Author Response · Authors · 2022-08-02
> **Authors response**
>
> We thank the reviewer for the valuable comments and will add the discussions below to our revised paper.
>
>
> > More qualitative comparisons and analysis on the non-rigid ray transformation
>
> We add more qualitative comparisons of the non-rigid ray transformation in Figure 2 in the link https://sites.google.com/view/totalselfscan. The results show that the ray transformation significantly improves the image quality. Without the ray transformation, there will be severe artifacts on the head, hands, and edge of the body. The reason is that the self-rotating video includes very limited human poses, and each point on the body is seen from a very limited range of view directions. Therefore, the color field network has difficulty producing high-quality rendering under novel human poses. Replacing the input view directions with global ones is a reasonable way to improve the generalization.
>
>
>
> > The prior of pose-dependent deformations
>
> To model pose-dependent deformations, we may leverage the existing multi-view human motion datasets to learn a generalizable pose-dependent deformation regressor, which can be conditioned on the human pose and canonical human geometry feature. Once learned, the generalizable regressor can be applied to new input data or can be further finetuned on the input data to improve the results.

---

> > ### Comment · Reviewer_Nipv · 2022-08-07
> > **Response to Author Feedback**
> >
> > Thanks for the authors’ effort in the feedback. All my concerns have been answered. I would recommend accepting this work.

---

### Official Review · Reviewer_LGtr · 2022-07-11

**Rating:** 6
**Confidence:** 4
**Soundness:** 3 good
**Presentation:** 3 good
**Contribution:** 2 fair

**Summary:**

This paper proposes to reconstruct the human body with detailed hands and face, leveraging separately recorded hand and face videos.

**Questions:**

Please see the Weaknesses section.

**Limitations:**

No additional limitations or potential negative societal impact.

**Strengths And Weaknesses:**

Strengths
1. The proposed pipeline is sound. Leveraging separate videos for body, face and hands at different scales makes sense; the fusion of geometry and appearance is well designed; the use of non-rigid ray transformation technique for free-view novel pose synthesis is reasonable.
2. The experiment results show more visually appealing results.
3. An additional SynTotalHuman dataset is proposed.

Weaknesses
My major concern is related to the use of parametric models as the foundation for reconstruction that undermines the technical significance of the paper. The reconstructed hands are claimed to be more refined than prior works, given that SMPL+H is used. Even without any replacement estimation, the SMPL+H already gives good geometry. This is particularly questionable as the examples shown all have bare hands.

---

> ### Author Response · Authors · 2022-08-02
> **Authors response**
>
> We thank the reviewer for the valuable comments and will add the discussions below to our revised paper.
>
>
>
> > The use of parametric models
>
> The use of parametric models indeed makes the problem easier but will not harm the applicability of the proposed method as we focus on human body reconstruction.
>
> The SMPL+H model gives reasonable hand geometry but it is quite coarse and lacks details. In Figure 1 in the link https://sites.google.com/view/totalselfscan, we show the comparison of hand geometry from the SMPL+H model and our reconstructed model. The results show that our method produces more detailed hand geometry such as bones and palm print.

---

### Official Review · Reviewer_3QzD · 2022-07-13

**Rating:** 7
**Confidence:** 4
**Soundness:** 3 good
**Presentation:** 3 good
**Contribution:** 4 excellent

**Summary:**

The paper presents a new method that creates full-body avatars from self-portrait monocular videos. The proposed method specifically models the body, the head, and the hands, the neural volumetric renderings of which are later fused together. Qualitative and quantitative evaluations demonstrate the superiority of the proposed method.

**Questions:**

It would be great if the authors could address the above mentioned points in the weaknesses section.

**Limitations:**

There are no potential negative societal impact of this paper.

**Strengths And Weaknesses:**

Paper strengths:
- The proposed new method demonstrates clear progress of neural human avatars, achieving sharp and clear renderings under novel views and novel poses.
- The proposed method is compared against strong existing baselines and ablated carefully.

Paper weaknesses:
- Despite the FLAME model being used for the head, its potential does not seem to be fully utilized. The facial expression remains static in the final results which significantly increases the uncanniness.
- The method description could benefit from more additional details. For instance, the latent code is mentioned several times but there lacks a description of it in the overall diagram. The $L^{body}$ loss was not defined in the main text.
- The non-rigid ray transformation seems to make the texture static across view directions. Was this the reason why the results seem quite rigid? Also, how does the proposed method perform when compared with a traditional graphics pipeline such as deformation transfer?

---

> ### Author Response · Authors · 2022-08-02
> **Authors response**
>
> We thank the reviewer for the valuable comments and will add the discussions below to our revised paper.
>
> > Potential of the FLAME model
>
>
> We thank the reviewer for the suggestion. As shown in RigNeRF [1] and IMavatar [2], the FLAME model can provide fine-grained face animation, given data with sufficient facial expressions. However, the data we currently collect does not contain rich expressions. In future work, we will collect the corresponding data and try to animate the face using the FLAME model.
>
>
>
>
> > More additional details of the method
>
> We will add more details about the method and update the overall diagram in the revised paper. The $L^{body}$ loss equals the $L^p$ loss in Equation (10) when the part $p$ denotes the body.
>
>
> > Non-rigid ray transformation
>
> To avoid unexpected artifacts under novel human poses, the non-rigid ray transformation makes the color unchanged with view direction, which sacrifices the view-dependent effect and makes the rendering less natural. Thus, there is a trade-off between better generalization to novel poses and more photo-realistic rendering.
>
>
> > Comparison with traditional graphics pipeline
>
> The traditional multi-view reconstruction methods cannot deal with the monocular self-portrait video in which the human is moving. A recent work VideoAvatar[3] proposed to deform a template body mesh to fit the image observations (e.g. silhouettes), but the reconstruction is less realistic due to the limited geometry accuracy and blurred texture map.
>
>
> ### References
>
> [1] Athar, ShahRukh, et al. "RigNeRF: Fully Controllable Neural 3D Portraits." CVPR 2022.
>
> [2] Zheng, Yufeng, et al. "Im avatar: Implicit morphable head avatars from videos." CVPR 2022.
>
> [3] Alldieck, Thiemo, et al. "Video based reconstruction of 3d people models." CVPR 2018.

---

### Meta-Review · Area_Chair_HuL9 · 2022-08-20

**Recommendation:** Accept
**Confidence:** Certain

**Metareview:**

This paper was reviewed by three experts in the field. Based on the reviewers' feedback, the decision is to recommend the paper for acceptance to NeurIPS 2022.

The reviewers did raise some valuable concerns that should be addressed in the final camera-ready version of the paper. For example, more discussion can be added on the key limitation of TotalSelfScan when applied to animation. The authors are encouraged to make the necessary changes to the best of their ability. We congratulate the authors on the acceptance of their paper!

**Award:**

No

---

### Decision · Program_Chairs · 2022-09-14

Accept